# A Novel Atomic-Level Post-Etch-Surface-Reinforcement Process for High-Performance *p*-GaN Gate HEMTs Fabrication

**DOI:** 10.3390/nano13162275

**Published:** 2023-08-08

**Authors:** Luyu Wang, Penghao Zhang, Kaiyue Zhu, Qiang Wang, Maolin Pan, Xin Sun, Ziqiang Huang, Kun Chen, Yannan Yang, Xinling Xie, Hai Huang, Xin Hu, Saisheng Xu, Chunlei Wu, Chen Wang, Min Xu, David Wei Zhang

**Affiliations:** 1State Key Laboratory of ASIC and System, Shanghai Institute of Intelligent Electronics & Systems, School of Microelectronics, Fudan University, Shanghai 200433, China; 2Imperial College London, London SW7 2AZ, UK; 3Shanghai Integrated Circuit Manufacturing Innovation Center Co., Ltd., Shanghai 200433, China

**Keywords:** etch induced damage, surface reinforcement, interface state, *p*-GaN gate HEMTs

## Abstract

A novel atomic-level post-etch-surface-reinforcement (PESR) process is developed to recover the *p*-GaN etching induced damage region for high performance *p*-GaN gate HEMTs fabrication. This process is composed of a self-limited surface modification step with O_2_ plasma, following by an oxide removal step with BCl_3_ plasma. With PESR process, the AlGaN surface morphology after *p*-GaN etching was comparable to the as-epitaxial level by AFM characterization, and the AlGaN lattice crystallization was also recovered which was measured in a confocal Raman system. The electrical measurement further confirmed the significant improvement of AlGaN surface quality, with one-order of magnitude lower surface leakage in a metal-semiconductor (MS) Schottky-diode and 6 times lower interface density of states (*D*_it_) in a MIS C-V characterization. The XPS analysis of Al_2_O_3_/AlGaN showed that the *p*-GaN etching induced F-byproduct and Ga-oxide was well removed and suppressed by PESR process. Finally, the developed PESR process was successfully integrated in *p*-GaN gate HEMTs fabrication, and the device performance was significantly enhanced with ~20% lower of on-resistance and ~25% less of current collapse at V_ds,Q_ bias of 40 V, showing great potential of leverage *p*-GaN gate HEMTs reliability.

## 1. Introduction

Compared with traditional silicon-based devices, gallium nitride (GaN) high electron mobility transistors (HEMTs) have high breakdown field, high carrier concentration and high electron mobility [1,2,3]. Considering cost and safe operation, the normally-off GaN-based HEMTs are more desirable in practical applications. Several approaches have been proposed to realize enhance mode operation, such as fluorine-plasma ion implantation [4], recessed gate [5] and *p*-GaN gate [6,7,8]. Among them, *p*-GaN gate HEMTs is the most promising candidate due to its excellent figure of merits and robust normally-off operation [9,10]. In principle, the conduction band of the AlGaN/GaN at the 2DEG channel is lifted up through the *p*-GaN gate, resulting in a normally-off operation with a positive threshold voltage [8]. However, selective removal of *p*-GaN and minimizing etch damage on the underlying AlGaN barrier layer are crucial in the access region for device performance [11].

The precise control of *p*-GaN etch is a key factor in *p*-GaN gate HEMTs fabrication. Outstanding etching depth control is imperative because the residual *p*-GaN layer in the out-of-gate area makes the 2DEG depleted and leads to a low output current. If the *p*-GaN layer is over etched into the AlGaN barrier layer, the 2DEG is also degraded because of the decrease in spontaneous polarization [12]. A highly selective ICP etching of *p*-GaN over AlGaN with BCl_3_/SF_6_ chemistry was systematically studies in our previous work, which features a well etching self-termination on the AlGaN barrier surface [13]. Nevertheless, further research revealed that etching-induced surface damage still exists on the underlying AlGaN surface [14], which could degrade the *p*-GaN gate HEMTs performance.

In this work, we report an atomic-level post-etch-surface-reinforcement (PESR) process to recover the damaged AlGaN surface. Both material and electrical characterization have been applied to demonstrate the effect of our developed PESR process to improve *p*-GaN gate HEMTs performance. 

## 2. Device Structure and Fabrication

For the fabrication of *p*-GaN gate HEMTs, the epitaxial structure was grown on a 6-inch Si (111) wafer by metal–organic chemical vapor deposition. The epitaxial III-nitride layers were composed of a 3.9 μm C-doped (Al)GaN buffer layer, a 200 nm GaN channel, a 12 nm Al_0.22_Ga_0.78_N barrier layer, a 0.8 nm AlN etch stop layer, and a 90 nm *p*-GaN layer. The density and mobility of holes in *p*-GaN were 2 × 10^17^ cm^−3^ and 15 cm^2^/Vs, respectively, by Hall measurement at room temperature.

Figure 1 shows the process schematic of the *p*-GaN gate HEMTs fabrication. After isolating the devices by BCl_3_/Ar deep etch, the *p*-GaN layer was selective etched by inductively coupled plasma (ICP) with BCl_3_/SF_6_. The F radicals from the SF_6_ plasma and Al atoms from the 0.8 nm AlN insert layer created a fluorination reaction and formed a thin aluminum fluoride (AlF_x_) etch-stop layer [15], resulting a highly selective etching process in the *p*-GaN/Al_0.22_GaN system. Subsequently, the formed AlF_x_ and the etching-induced surface damage was removed by the atomic-level PESR process, which combines self-limiting surface O-modification and oxide removal steps based on the customized ultra-low power ICP equipment. As shown in Table 1, the etched layer was oxidized by oxygen plasma treatment at the first step, and the so-formed oxide was selectively removed using low energy BCl_3_ plasma, which had negligible effect on the un-oxidized AlGaN material. Then, a 15 nm Al_2_O_3_ passivation layer was deposited at 300 °C by ALD. The source/drain ohmic contacts were formed by a lift-off process of e-beam-evaporated Ti/Al/Ni/Au metal layers and rapid thermal annealing process. PVD TiN metal was used as the gate electrode. In addition, transmission line model (TLM) for ohmic contact test analysis and MIS diode structure for capacitance–voltage (*C-V*) tests were also fabricated on the same wafer.

## 3. Results and Discussion

For the as-epitaxial AlGaN surface, the RMS was about 0.42 nm. After *p*-GaN selective etching, RMS was increased to ~0.89 nm, which was resulted by large numbers of nanopillars on the etched surface as shown in Figure 2a. The developed PESR process could effectively modify the etch-induced damage surface because of the self-limiting properties of O_2_ plasma oxidation and the following etch step with BCl_3_ chemistry. 

Further surface characterization was performed by Raman under 405 nm excitation laser wavelength at room temperature, as shown in Figure 2b. Compared with the as-grown AlGaN/GaN heterostructure, the AlGaN A_1_(LO) peak was shifted negatively [16] after *p*-GaN etch as seen in Figure 2b middle, which indicated near surface lattice damage during the *p*-GaN etch process. Amazingly as seen in Figure 2b bottom, the AlGaN A_1_(LO) peak moved back to the as-epitaxial position, which demonstrated the effectiveness of removing the damaged surface with the atomic-level PESR treatment.

Schottky diodes were fabricated on etched AlGaN to characterize the surface quality. As shown in Figure 2c, with the application of PESR technology, the MS Schottky leakage current was reduced by an order of magnitude at 10 V bias. This strongly indicated that the developed PESR process could recover the etched AlGaN surface by self-limiting removing the byproducts or damaged layer after *p*-GaN etching. Specific contact resistivity (*R*_c_) and sheet resistance measurements (*R*_sheet_) were made using a test pattern conforming to the linear transmission line model (TLM) as described by Reeves et al. [17]. The TLM pattern consists of 100 × 100 μm contact pads with separations of 7 μm, 10 μm, 15 μm, 20 μm, 40 μm, 60 μm. *R*_sheet_ and *R*_c_ are extracted from the slope and the *y*-axis intercept of their corresponding linear fits, respectively. As shown in Figure 2d, comparing with as etched sample, the sheet resistance was reduced by 69 Ω/□ after PESR, which indicated that 2DEG transport characteristics was improved.

Figure 3a,b show the multi-frequency *C-V* curves of the TiN/Al_2_O_3_/AlGaN/GaN MIS capacitors. The *C-V* hysteresis at 1 MHz was reduced from ~225 mV to ~115 mV after PESR treatment, indicating that there was less electron trapping in the Al_2_O_3_/AlGaN MIS structure. Al_2_O_3_/AlGaN interface state density (*D*_it_) was derived from *C-V* frequency dispersion [18]. For MIS *C-V* with PESR process, the measured Δ*V*_FB_ between 100 kHz and 1 MHz was ~180 mV, giving *D*_it_ of 7.20 × 10^12^ cm^−2^ eV^−1^ with time constant in the range from 0.16 μs to 1.6 μs. The corresponding *D*_it_ for as-etched Al_2_O_3_/AlGaN MIS capacitor was as high as 4.62 × 10^13^ cm^−2^ eV^−1^. The decrease of trap states may be attributed to the effect of the beneficial of interface reinforcement. It can be seen that the result in Figure 3b has a less frequency hysteresis and a better high frequency performance.

In order to further understand the mechanism of PESR process, both XPS and TEM characterization were performed and analyzed. After normal *p*-GaN etch, the surface presented a high intensity of F-bond in Figure 4a, indicating a possible amorphous fluoride layer at Al_2_O_3_/AlGaN interface. In contrast, the PESR process significantly got rid of the F-bond layer as seen in Figure 4a. Meanwhile, the Ga-O bonds and the native oxides were also effectively suppressed, resulting in a sharper interface between Al_2_O_3_ and AlGaN as shown in Figure 4b,c. The whole process was illustrated in the schematic diagram as in Figure 4d–f. The etch-induced byproducts layer was removed and the damaged surface was modified after PESR process, thus generated a better AlGaN surface for later-on ALD-Al_2_O_3_ deposition.

Figure 5a shows the *p*-GaN gate HEMTs transfer characteristics at *V*_ds_ = 10 V of as-etched and after PESR, respectively. A normally-off operation with a *V*_th_ of 1.1 V is achieved. As seen in log-scale, due to the improved Al_2_O_3_/AlGaN interface in drift region with PESR process, the device gate leakage *I*_g_ was reduced by an order of magnitude, leading to as high as 6 × 10^9^ on-off ratio. The saturated drain current (*I*_sat_) of device after PESR is 325 mA/mm at *V*_g_ = 5 V, which is about 1.2 times the 270 mA/mm of as-etched one, as illustrated in Figure 5b. The extracted on-resistance (*R*_ON_) of as-etched device and after PESR one are 14.3 Ω⋅mm and 11.8 Ω⋅mm at *V*_g_ = 5 V, respectively.

Pulsed *I*_ds_-*V*_ds_ measurements under slow switching were performed to characterize dynamic ON-resistance (*R*_ON,D_) of the fabricated *p*-GaN gate HEMTs [19]. The dynamic characteristics of the fabricated *p*-GaN gate HEMTs were investigated using the Keithley 4200A PMU module with the drain static bias voltage *V*_ds,Q_. The period of the square wave pulse signal is 1 ms, the width is 10 μs, the duty cycle is 1%, and the time of pulse rising and falling is set to 500 ns. The device is synchronously switched from an OFF-state quiescent bias of *V*_gs,Q_ = −5 V, *V*_ds,Q_ = 0/10/20/30/40 V to measurement state of *V*_gs,M_ = 5 V and *V*_ds_ from 0 V to 10 V. Figure 6a,b illustrate the current collapse phenomenon under increasing *V*_ds,Q_, which was obviously suppressed after PESR process, indicating that Al_2_O_3_/AlGaN interface states was improved. Meanwhile, the ratio of *R*_ON,D_/*R*_ON,S_ for as-etched device was 1.45, which was significantly improved to 1.19 for devices with PESR process at *V*_gs,Q_ = 5 V, *V*_ds,Q_ = 40 V as shown in Figure 6d.

## 4. Conclusions

In summary, a novel atomic-level post-etch-surface-reinforcement (PESR) process is developed to recover the *p*-GaN etching induced damage region for high performance *p*-GaN gate HEMTs fabrication. This process is composed of a surface modification step with O_2_ plasma, following by an oxide removal step with BCl_3_ plasma, which made negligible damage on the un-oxidized AlGaN to obtain a high-quality AlGaN surface due to the self-limiting characteristic. As a result, the fabricated HEMTs device performance was significantly enhanced with ~20% lower of on-resistance, and ~25% less of current collapse at *V*_ds,Q_ bias of 40 V with PESR process, showing great potential of leverage *p*-GaN gate HEMTs reliability.

## Figures and Tables

**Figure 1 nanomaterials-13-02275-f001:**
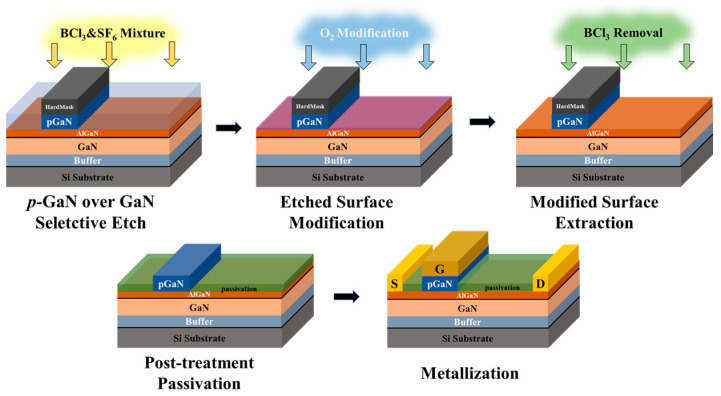
Schematic process flow of the *p*-GaN gate HEMTs fabrication.

**Figure 2 nanomaterials-13-02275-f002:**
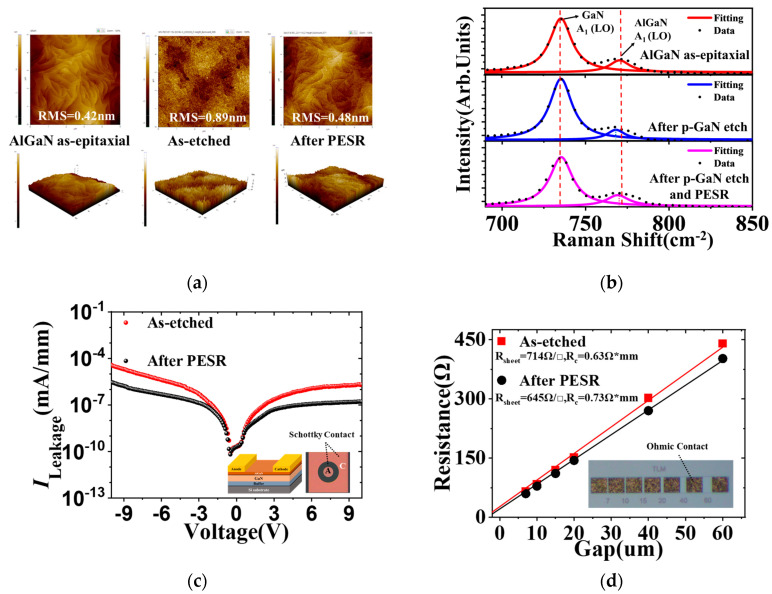
(**a**) 5 × 5 μm^2^ AFM images; (**b**) The Raman spectra of AlGaN/GaN heterostructure of as-epitaxial AlGaN surface, after *p*-GaN etch and PESR, after normal *p*-GaN etch; (**c**) Leakage current in a Metal-Semiconductor (MS) Schottky Diode; (**d**) *I-V* measurements and linear fit for the TLM. The inset shows the TLM patterns.

**Figure 3 nanomaterials-13-02275-f003:**
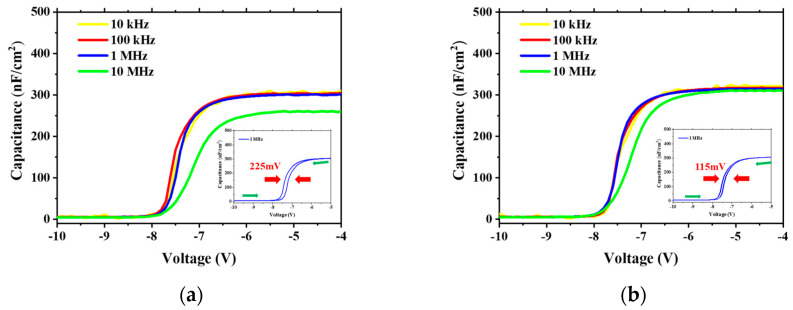
Multi-frequency *C-V* curves for MIS capacitors of (**a**) after normal *p*-GaN etch, and (**b**) after *p*-GaN etch and with PESR. The insets were the hysteresis curves at 1 MHz.

**Figure 4 nanomaterials-13-02275-f004:**
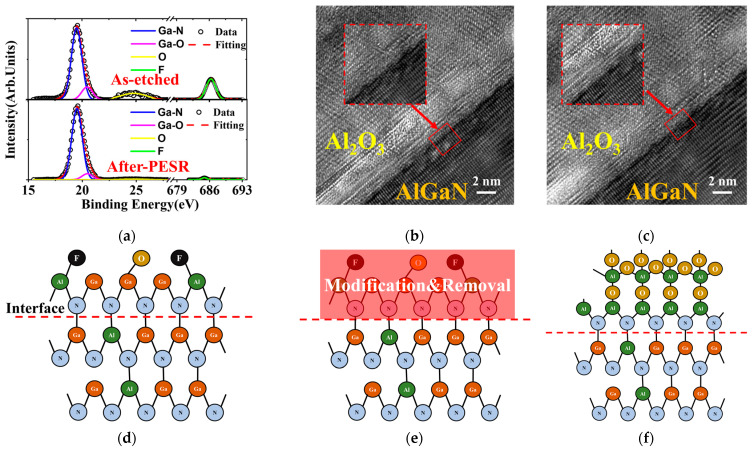
XPS spectra of Al_2_O_3_/AlGaN interface (**a**); TEM of Al_2_O_3_/AlGaN interface (**b**) after normal *p*-GaN etch and (**c**) after *p*-GaN etch and PESR; The schematic diagram of atomic level PESR process (**d**–**f**).

**Figure 5 nanomaterials-13-02275-f005:**
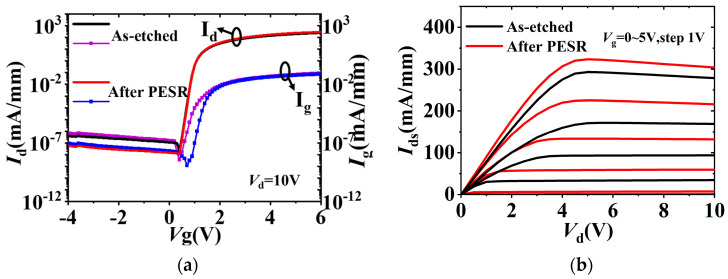
(**a**) Transfer characteristics in semi-logarithm scale, (**b**) output characteristics of the fabricated *p*-GaN gate HEMTs.

**Figure 6 nanomaterials-13-02275-f006:**
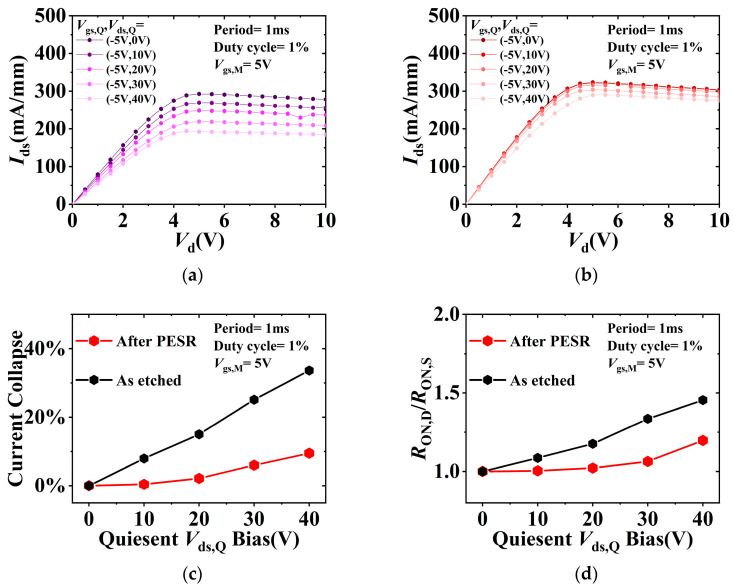
Pulsed *I*_ds_-*V*_ds_ characteristics under different quiescent biases (**a**) as-etched and (**b**) after PESR; (**c**) current collapse and (**d**) ratio of the dynamic *R*_ON,D_/*R*_ON,S_ of the fabricated *p*-GaN gate HEMTs.

**Table 1 nanomaterials-13-02275-t001:** The atomic-level PESR process condition.

Parameter	O_2_ Modification	BCl_3_ Removal
ICP power (W)	400	200
Bias power (W)	0	6
Chamber pressure (mTorr)	10	10
Gas flow rate (sccm)	100	100
Treatment time (s)	15	5

## Data Availability

Not applicable.

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
