# Peer review of "A Novel Atomic-Level Post-Etch-Surface-Reinforcement Process for High-Performance p-GaN Gate HEMTs Fabrication"

_nanomaterials, 2023, doi:10.3390/nano13162275_

Round 1

Reviewer 1 Report

After reading carefully the manuscript "A novel atomic-level post-etch-surface-reinforcement process for high-performance p-GaN Gate HEMTs fabrication" proposed by L. Wang et al., where a method composed by surface modification with a O2 plasma and the oxide is removed using a BCl3 plasma with small effect on the AlGaN is proposed, I concluded that the manuscript is appropriate for the Nanomaterials journal. 

The article is very clear and well written and the ideas are perfectly described. I just recommend the authors to improve some of the images. for example, in image 2 (AFM) the authors included the RMS roughness but the units in 0.42, 0.89 and 0.48 are missing. I believe it should be nanometers.

Author Response

We really appreciate your favorite consideration and insightful comments concerning our manuscript entitled “A Novel Atomic-Level Post-Etch-Surface-Reinforcement Process for High-Performance p-GaN Gate HEMTs Fabrication”. Those comments are very valuable and helpful for improving the quality and readability of our paper. We have studied the comments carefully and revised the paper accordingly as below.

Point 1: The article is very clear and well written and the ideas are perfectly described. I just recommend the authors to improve some of the images. for example, in image 2 (AFM) the authors included the RMS roughness but the units in 0.42, 0.89 and 0.48 are missing. I believe it should be nanometers.

Response 1: Thank you very much for your good suggestion. The units (nm) of the RMS roughness has been added in Figure 2 (a).

Corresponding change in manuscript: Yes.

Location of Change: Page 3 ,Figure 2 (a).

Reviewer 2 Report

Luyu Wang, et al. report the experimental study of a post-etch-surface-reinforcement process to recover the p-GaN etching induced damage region for high performance p-GaN gate HEMTs fabrication. A rather important technological result is obtained, consisting in a significant improvement of the AlGaN surface quality, with an order of magnitude decrease in the surface leakage in the metal-semiconductor Schottky diode. They developed an etching model and showed that p-GaN etching produces a fluoride product that is well removed and suppressed by the PESR process and finally integrated into the HEMT gate. The results could lead to interesting applications of the the HEMT gate and well fitted with the scope of Nanomaterials.

I have a few smaller comments/questions about the experimental results and analysis, which are listed below.

Did the authors measure and observe any changes in the XPS spectra of nitrogen upon chemical treatment?  Is the behavior of the nitrogen line consistent with the conclusions drawn?

I would also encourage the authors to look at the wet chemical treatment developed in [Solid State Physics, Vol. 46, No. 10, 2004, p. 1949], which allows the preparation of a well-ordered p-GaN surface.

Author Response

We really appreciate your favorite consideration and insightful comments concerning our manuscript entitled “A Novel Atomic-Level Post-Etch-Surface-Reinforcement Process for High-Performance p-GaN Gate HEMTs Fabrication”. Those comments are very valuable and helpful for improving the quality and readability of our paper. We have studied the comments carefully and revised the paper.
